# Structural definition of a pan-sarbecovirus neutralizing epitope on the spike S2 subunit

Nicholas K. Hurlburt [1,8], Leah J. Homad [1,8], Irika Sinha [1], Madeleine F. Jennewein[1], Anna J. MacCamy[1], Yu-Hsin Wan[1], Jim Boonyaratanakornkit [1], Anton M. Sholukh [1], Abigail M. Jackson [2], Panpan Zhou [3], Dennis R. Burton [3,4], Raiees Andrabi [3], Gabriel Ozorowski [2], Andrew B. Ward [2], Leonidas Stamatatos [1,5✉], Marie Pancera [1,6✉] & Andrew T. McGuire [1,5,7✉]

Three betacoronaviruses have crossed the species barrier and established human-to-human transmission causing significant morbidity and mortality in the past 20 years. The most current and widespread of these is SARS-CoV-2. The identification of CoVs with zoonotic potential in animal reservoirs suggests that additional outbreaks could occur. Monoclonal antibodies targeting conserved neutralizing epitopes on diverse CoVs can form the basis for prophylaxis and therapeutic treatments and enable the design of vaccines aimed at providing pan-CoV protection. We previously identified a neutralizing monoclonal antibody, CV3-25 that binds to the SARS-CoV-2 spike, neutralizes the SARS-CoV-2 Beta variant comparably to the ancestral Wuhan Hu-1 strain, cross neutralizes SARS-CoV-1 and binds to recombinant proteins derived from the spike-ectodomains of HCoV-OC43 and HCoV-HKU1. Here, we show that the neutralizing activity of CV3-25 is maintained against the Alpha, Delta, Gamma and Omicron variants of concern as well as a SARS-CoV-like bat coronavirus with zoonotic potential by binding to a conserved linear peptide in the stem-helix region. Negative stain electron microscopy and a 1.74 Å crystal structure of a CV3-25/peptide complex demonstrates that CV3-25 binds to the base of the stem helix at the HR2 boundary to an epitope that is distinct from other stem-helix directed neutralizing mAbs.

[1] Fred Hutchinson Cancer Research Center, Vaccine and Infectious Disease Division, Seattle, WA, USA. [2] Department of Integrative Structural and Computational Biology, The Scripps Research Institute, La Jolla, CA, USA. [3] Department of Immunology and Microbiology, The Scripps Research Institute, La Jolla, CA, USA. [4] Ragon Institute of MGH, MIT and Harvard, Cambridge, MA, USA. [5] Department of Global Health, University of Washington, Seattle, WA, USA. [6] Vaccine Research Center, NAID, NIH, Bethesda, MD, USA. [7] Department of Laboratory Medicine and Pathology, University of Washington, Seattle, WA, USA. [8] These authors contributed equally: Nicholas K. Hurlburt, Leah J. Homad. ✉email: lstamata@fredhutch.org; mpancera@fredhutch.org; amcguire@fredhutch.org

Coronaviruses (CoVs) are a large family of viruses that infect many species of birds and mammals, including humans. They are subdivided into four genera; alpha, beta, gamma and delta. Two alpha-CoVs, NL63 and 229E, and two beta-CoVs (OC43 and HKU1) are endemic in the human population and cause mild respiratory cold like symptoms[1]. Three separate zoonotic transmissions of highly pathogenic beta-CoVs to humans have been documented in the last two decades.

Middle East respiratory syndrome coronavirus (MERS-CoV) first emerged in Saudi Arabia in 2012 and has since been detected in 27 countries (Zaki et al., 2012). There have been ~2574 reported MERS cases resulting in 884 deaths (35.4% mortality rate).

SARS-CoV-1 was first identified as the causative agent of atypical respiratory syndrome called Severe Acute Respiratory Syndrome in China in 2002. SARS-CoV-1 infected 8098 people causing 774 deaths (9.5% mortality rate). More recently, the highly transmissible SARS-CoV-2 virus emerged in China and rapidly spread through the global population. SARS-CoV-2 has infected ~185 million people and caused over 4 million deaths[2]. SARS-CoV-2 and SARS-CoV-1 are members of the sarbecovirus subgenus and share ~80% amino acid sequence identity[3]. SARS-CoV-2 is highly similar to a bat CoV, RaTG13 (Zhou et al., 2020). Several other SARS-like bat coronaviruses have been identified that have zoonotic potential[1] suggesting that both viruses likely originated in bats.

CoV infection is mediated by the viral spike protein (S) which is a membrane anchored class I fusion protein expressed on the virion surface and is an important target of host immune responses elicited by infection or vaccination. S is comprised of two distinct functional subunits; a N-terminal, membrane distal subunit designated S1 and a C-terminal, membrane proximal subunit designated S2. The S2 domain houses the fusion machinery that undergoes large structural rearrangements to mediate fusion of the host and viral membranes. The S1 domain serves to stabilize the S2 subunit in the pre-fusion state and facilitates the attachment to ligands on host cells through the receptor binding domain (RBD)[4]. In general, CoV-cell fusion requires conformational changes induced by receptor binding, as well as further proteolytic cleavage of the S2 subunit to liberate the fusion peptide and trigger conformational changes, which can occur at the cell membrane or following viral endocytosis[5]. The SARS-CoV-2 spike is translated as a single polypeptide that is proteolytically cleaved at a furin site between the S1 and S2 subunits in the virus-producer cell[6,7]. Following binding to the ACE2 receptor on the target cell, cleavage at the S2' site by TMPRSS at the cell surface or cathepsin L, following endocytosis is required to liberate the fusion peptide[7–11]. Rearrangements of S2 embed the fusion peptide into the host membrane and then refolding results in the formation of a fusion pore[12,13].

Despite the overall structural similarity of their S proteins, human coronaviruses use diverse entry receptors[4,7,14]. 229E uses human aminopeptidase N (hAPN), while HKU1 and OC43 cell-entry depends on 9-O-acetylated sialic acids[4,15,16]. NL63, SARS-CoV-1 and SARS-CoV-2 use angiotensin converting enzyme 2 (ACE2)[7,8,17,18]. SARS-CoV-2 also uses heparan sulfate as an attachment factor to promote infection[19]. MERS-CoV utilizes sialoside receptors as attachment factors and dipeptidyl peptidase 4 (DPP4) as an entry receptor[20,21].

Due to extensive CoV genetic diversity, wide range of animal hosts, and potential for zoonotic transmission there is a need for vaccines and therapeutic agents that can prevent or limit future outbreaks[22,23]. Neutralizing antibodies elicited by vaccination or natural infection are an important correlate of protection against subsequent CoV infection[24–26]. Further, passive delivery of neutralizing monoclonal antibodies (mAbs) can be used as a countermeasure to prevent CoV-related illness[27].

The primary targets of neutralizing antibodies are within the S1 subunit: the receptor binding domain (RBD) and the N-terminal domain (NTD)[28–47]. Due to the diversity in receptor usage and variability of spike sequences across CoVs, RBD and NTD directed mAbs are often virus specific. Even within the same CoV, mutant variants can evade neutralization by mAbs and polyclonal sera. Indeed, mutations found in the RBD and NTD of SARS-CoV-2 variants of concern are responsible for increased resistance to serum and mAbs. It has been speculated such variants could erode vaccine efficacy over time. The RBD and NTD of other CoVs appear to be subject to and evade immune pressure as well[44,45]. In contrast, S2 is more functionally and structurally conserved among CoVs[4,48]. However, it is sub-dominant with respect to neutralizing antibody responses as the majority of S2-binding mAbs isolated from SARS-CoV-2 infected donors are non-neutralizing[32,33,49–51].

We recently described the isolation of a neutralizing anti-S2 mAb, CV3-25 from a SARS-CoV-2 infected donor[50]. The neutralizing potency of CV3-25 is unaffected by mutations found in the Beta (B.1.351) SARS-CoV-2 variant and can neutralize SARS-CoV-1. CV3-25 also displays cross-reactivity with recombinant spike proteins derived from the betaCoVs, OC43 and HKU1. Here, we demonstrate that the neutralizing activity of CV3-25 is unaffected by mutations found in the Alpha[52], Delta[53], Gamma[54], and Omicron[55] variants and show that it can neutralize a sarbecovirus from bats, WIV1[56]. We identified a linear epitope overlapping the stem-helix/HR2 region containing the epitope of CV3-25. A crystal structure of CV3-25 with a 19mer peptide revealed that CV3-25 binds to a solvent-exposed linear epitope that partially unwinds the stem-helix. The CV3-25 epitope is distinct from other mAbs targeting the stem helix region[57–59], thus defining a site of conserved vulnerability that will enable pan-CoV vaccine design.

## Results

### CV3-25 neutralizes SARS-CoV-2 variants and a SARS-like bat coronavirus.
We previously reported that CV3-25 neutralizes the Wuhan-Hu-1 and Beta (B.1.351) variants of SARS-CoV-2 with comparable potency in a pseudovirus neutralization assay[50]. Here we evaluated the ability of CV3-25 to neutralize additional SARS-CoV-2 variants Alpha (B.1.1.7), Delta (B.1.617.2), Gamma (P.1), and Omicron (B.1.1.529) and a more distantly related sarbecovirus from bats, WIV1, which uses ACE2 as an entry receptor and can infect human cell lines[52,54,56,60]. Therefore, WIV1 represents a bat CoV with pandemic potential. CV3-25 neutralized all variants and WIV1 with comparable potency (Fig. 1a and Supplementary Table 1). In contrast, the RBD-directed CV30 mAb showed reduced potency against the Beta, and Gamma variants of concern both of which harbor mutations in the RBD at position 417 that makes direct contact with CV30[54,60,61] (Fig. 1b and Supplementary Table 1). CV30 was unable to neutralize the Omicron variant which harbors mutations at 477 and 493 in the RBD that are contact residues for CV30[61]. WIV1 was completely resistant to CV30-mediated neutralization. None of the pseudoviruses were neutralized by the anti-EBV mAb AMMO1 (Fig. 1c)[62]. Combined with the observation that CV3-25 also neutralizes SARS-CoV-1, these data indicate that it binds to an epitope on S2 that is unaffected by mutations found in these sarbecovirus variants.

### CV3-25 binds to a linear epitope on the SARS-CoV-2 stem helix.
Due to the ability of CV3-25 to neutralize diverse sarbecoviruses including SARS CoV-1, SARS-CoV-2 and WIV1 (Fig. 1), and the fact that it can bind the ectodomains from additional betacoronaviruses, OC43, and HKU1[50], we sought to

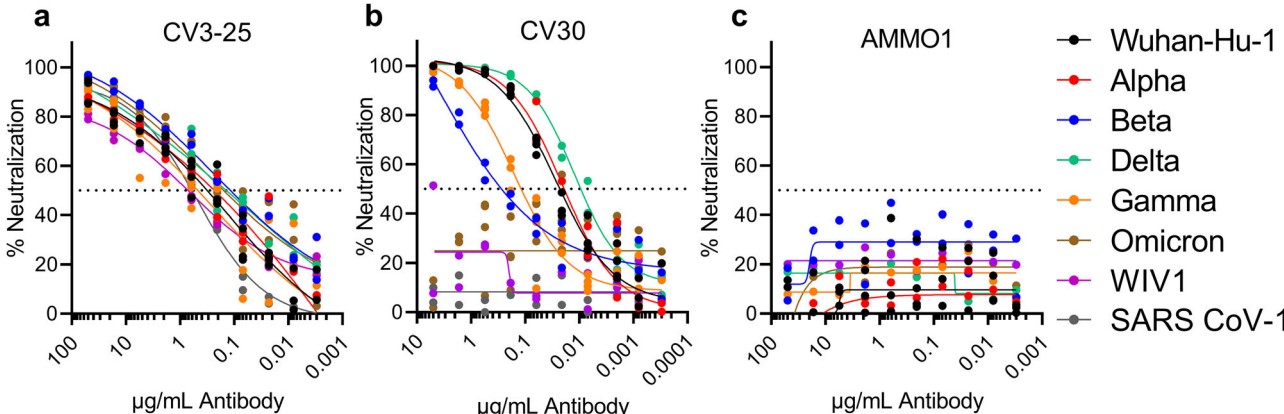

**Fig. 1 CV3-25 neutralizes SARS-CoV-2 variants and a SARS-like Bat virus.** The S2-binding CV3-25 (**a**), RBD-binding CV30 (**b**) and anti-EBV mAb AMMO1 (**c**) were evaluated for their ability to neutralize the indicated SARS-CoV-2 variants of concern and the SARS-like bat virus WIV1 in a pseudovirus assay. Each dot represents a technical replicate from one or two independent experiments. Curve fits are shown as solid lines.

delineate the epitope targeted by CV3-25 in the context of the SARS-CoV-2 spike protein. To this end we carried out negative stain electron microscopy (nsEM) of CV3-25 IgG complexed with a stabilized spike ectodomain (S6P). 2D class averages indicated that CV3-25 binds to the base of the SARS-CoV-2 ectodomain (Fig. 2a). A 3D reconstruction of the CV3-25/S6P complex at ~33 Å resolution revealed that CV3-25 bound with an apparent stoichiometry of one Fab per trimer to the stem region of the spike (Fig. 2b and Supplementary Fig. 1).

Several mAbs have been found to bind the stem helix at the base of the S2 spike that display varying degrees of CoV cross-binding and/or neutralizing activity, and all are poorly resolved by negative stain or cryoEM[58,59,63–66]. This epitope region is conformationally dynamic and poorly resolved in unliganded EM reconstructions of the spike protein as well[7,13,14,67].

We performed binding competition studies with the B6 and CC40.8 stem-helix-directed mAbs to verify that CV3-25 is binding to this epitope region. B6 neutralizes pseudoviruses expressing the spike proteins from MERS-CoV, HCoV-OC43 and the MERS-like bat CoV, HKU4. It binds to but does not neutralize SARS-CoV-1 and SARS-CoV-2[57]. CC40.8 binds to an epitope that is nearly identical to B6 and weakly neutralizes SARS-CoV-1 and SARS-CoV-2[59,65]. B6 and CC40.8 readily bound to the S6P protein as measured by biolayer interferometry (BLI), but they showed greatly reduced binding to an S6P-CV3-25 complex indicating that the antibodies compete for binding to the SARS-CoV-2 spike (Fig. 2c, d). In contrast the binding of CV30, an RBD-directed mAb was largely unaffected by CV3-25 binding (Fig. 2e).

B6 and CC40.8 bind to linear peptides spanning amino acids 1147-1157, and 1142-1159 of the SARS-CoV-1/SARS-CoV-2 stem helix, respectively[57,59]. To test whether CV3-25 binds to a similar epitope, CV3-25 binding to overlapping 15mer linear peptides spanning the stem helix region (1143–1162) from SARS-CoV-2 was measured by ELISA (Fig. 2f). CV3-25 bound to two peptides encompassing amino acids 1149–1163 and 1153–1167, with stronger binding to the latter (Fig. 2g). CV3-25 did not bind to any of the other SARS-CoV-2 peptides or to a control peptide from HIV-1 Env. CV3-25 binding was specific, as CV2-10, an S2 directed mAb that does not compete with CV3-25 binding[50], did not bind either peptide (Fig. 2h). To confirm binding to this peptide region, we synthesized a peptide spanning 1145–1167 and verified that CV3-25 bound to the peptide with ~5 nM affinity using biolayer interferometry (BLI) (Fig. 2i and Supplementary Table 2). The measured affinity of CV3-25 for the peptide is lower than it is for a recombinant stabilized spike protein (~0.6 nM)

(Fig. 2j and Supplementary Table 2). We note that the association rate of binding to the peptide does not fit well to a 1:1 binding model (Fig. 2i) which may reflect several conformations sampled by the immobilized peptide (a heterogenous ligand) and affect the accuracy of the CV3-25 peptide binding measurement. Alternatively, the difference in affinity might be attributed to additional contacts made between CV3-25 and the spike protein outside of the linear peptide tested here.

**Structure of CV3-25 reveals a site of vulnerability in S2.** To gain insight into the nature of the CV3-25 peptide interaction, the antigen binding fragment (Fab) of CV3-25 was complexed with a synthesized peptide of the C-terminal end of the stem helix (residues 1149–1167). A crystal structure of the Fab-peptide complex was obtained to a resolution of 1.74 Å (Table 1). The structure showed that binding to this peptide is almost entirely heavy chain dependent (Fig. 3a, b). The N-terminal end of the peptide forms an α-helix that is engaged by the CDRH1 and CDRH2. The CDRH3 extends under the base of the α-helix directing the extended C-terminal portion of the peptide up into the CDRH1 before turning downward to interact with the light chain. The Fab binds the peptide with a total buried surface area (BSA) of ~594 Å², of which ~516 Å² is from the heavy chain and ~78 Å² from the light chain (Fig. 3c).

Alanine scanning of a stem helix peptide was conducted to assess the relative contributions of the interactions observed in the crystal structure (Fig. 3d and Supplementary Fig. 2). This analysis revealed that mutating $Lys_{1157}$, any of the residues in $_{1160}TSPDV_{1164}$, or $Leu_{1166}$ inhibited or greatly reduced binding (Fig. 3d). This data agrees well with the structural data. ~133 Å² of $Lys_{1157}$ is buried upon binding, the highest amount of BSA on the peptide, and forms hydrogen bonds with two Asp residues in the CDRH2 (Fig. 3e). $_{1160}TSPDV_{1164}$ is the segment of peptide just after the helix that interacts with CDRH3 before curving up to interact with CDRH1 and then the light chain.

Reversion of CV3-25 to the inferred germline (iGL) version abrogated CV3-25 neutralizing activity despite showing comparable binding to SARS-CoV-S2P under avid conditions[50]. Although the majority of the mAb-peptide contacts are through the CDRH3, $Arg_{31}$ in the CDRH1 has the highest buried surface area (75 Å²) upon binding the peptide (Fig. 3c). $Arg_{31}$ forms a water-mediated interaction with $Asp_{1153}$ and a π-stacking interaction with $Phe_{1156}$ on the peptide. The germline encoded Ser at this position would be incapable of forming these interactions providing a rationale for the lack of neutralizing activity of iGL-CV3-25.

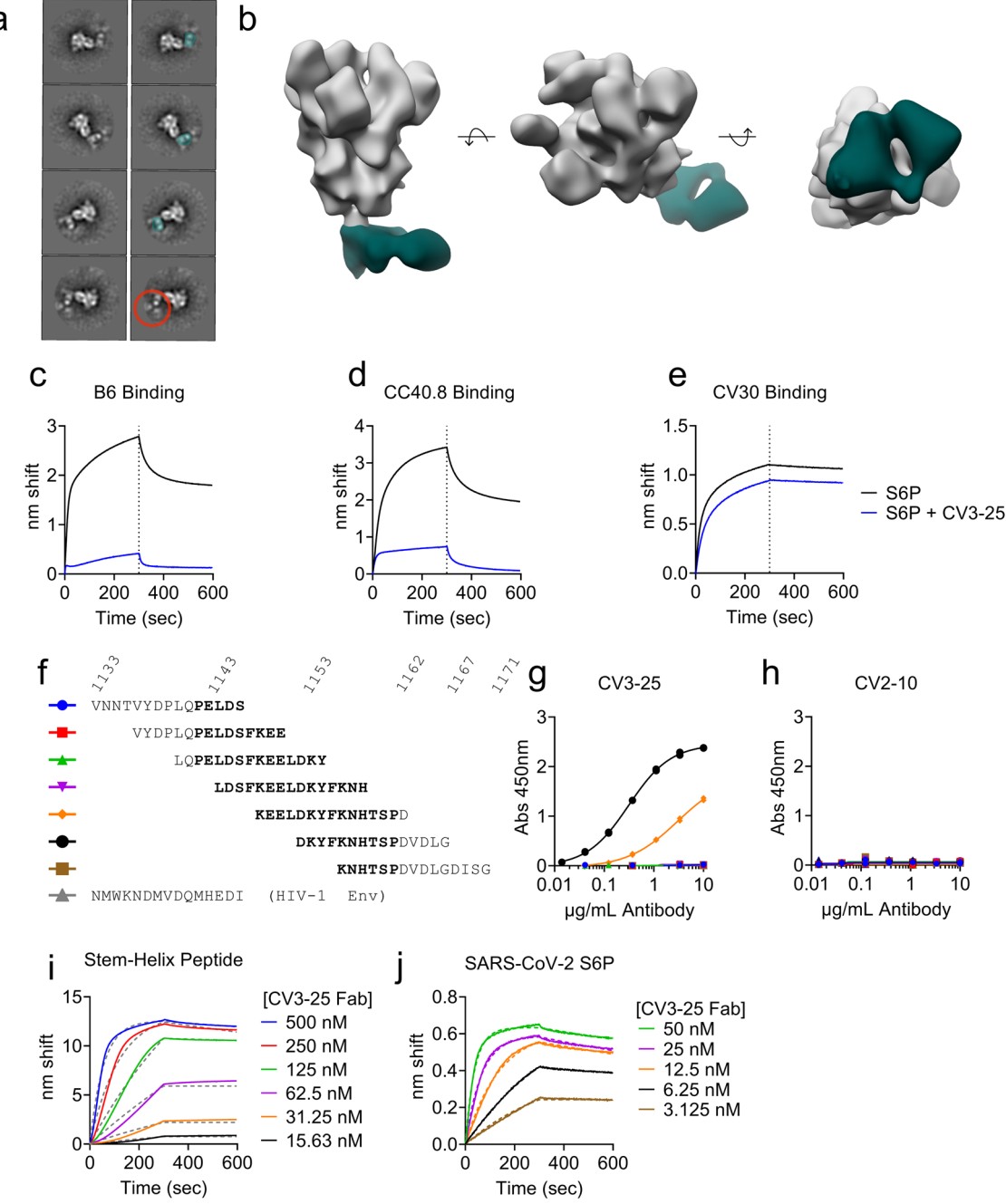

**Fig. 2 CV3-25 binds to a linear peptide encompassing the C-terminus of the stem helix. a** Representative 2D class averages of CV3-25 IgG bound to SARS-CoV-2 6P-D614G S protein by negative stain electron microscopy. A single Fab of the IgG—highlighted in teal—can be seen bound to the lower S2 domain of the S protein. The red circle highlights the fullly IgG visible in 2D. Images without highlights are shown in the left column for comparison. **b** Segmented 3D reconstruction of a CV3-25 Fab (dark green) bound to the lower S2 domain of the S protein. Binding of the B6 (**c**), CC40.8 (**d**), and CV30 (**e**) mAbs to SARS-CoV-2 S6P alone or a SARS-CoV-2 S6P-CV3-25 complex as indicated. **f** Alignment of a set of 15mer peptides that overlap by 11 amino acids spanning residues 1133–1171 of the SARS-CoV-2 spike protein. The region that corresponds to the stem helix in the prefusion wild-type spike protein (based on the 6XR8) is shown in bold. **g** CV3-25 was tested for binding to the peptides in (**e**), and to a 15mer peptide derived from an HIV-1 Env protein. **h** CV2-10, which also binds to S2, but does not compete with CV3-25 was tested for binding to the peptides in (**f**). Each data point represents a technical replicate conducted in duplicate in (**g**) and (**h**). CV3-25 Fab binding was measured to the SARS-CoV-2 stem helix peptide: 1145-LDSFKEELDKYFKNHTSPDVDLG-1167 (**i**) or a stabilized SARS-CoV-2 spike protein by BLI (**j**).

Structural alignment of the stem helix peptides in the CV3-25 and B6 structures show that CV3-25 binds more C-terminal than B6 in the stem helix (Fig. 4a) in agreement with the binding to overlapping linear peptides (Fig. 2e, f). The stem helix residues that are shared in the structures adopt almost identical conformations (Fig. 4a). B6 binds to the hydrophobic face of the

amphipathic helix that is predicted to be on the interior of the stem helix bundle of the pre-fusion trimer[57]. In contrast, CV3-25 binds to the solvent-exposed hydrophilic face of the helix. An alignment of the CV3-25 stem helix to a cryoEM structure of the native prefusion spike with the stem helix structure resolved (PDBID: 6XR8)[13] indicates that the CV3-25-bound stem helix

**Table 1 Data collection and refinement statistics for crystal structure.**

|  | CV3-25 Fab + Spike peptide 1149–1167 |
|---|---|
| **Data collection** |  |
| Space group | P3$_2$21 |
| Cell dimensions |  |
| $a$, $b$, $c$ (Å) | 60.173, 60.173, 285.825 |
| $\alpha$, $\beta$, $\gamma$ (°) | 90, 90, 120 |
| Resolution (Å) | 49.01–1.740 (1.77–1.74) |
| $R_{merge}$[a] | 0.025 (0.309) |
| <I/σ(I)> | 21.7 (2.7) |
| CC$_{1/2}$ | 0.999 (0.670) |
| Completeness | 100 (100) |
| Redundancy | 1.9 (1.9) |
| **Refinement** |  |
| Resolution (Å) | 48.96–1.74 (2.18–1.74) |
| No. unique reflections | 63190 (6196) |
| $R_{work}$[b]/$R_{free}$[c] | 18.64/20.90 (34.85/35.03) |
| No. atoms | 3825 |
| Protein | 3479 |
| Water | 294 |
| Ligand | 52 |
| B-factors (Å$^2$) | 36.1 |
| Protein | 35.24 |
| Water | 42.33 |
| Ligand | 58.63 |
| RMS bond length (Å) | 0.012 |
| RMS bond angle (°) | 1.57 |
| **Ramachadran Plot Statistics**[d] |  |
| Residues | 455 |
| Most Favored region | 97.54 |
| Allowed Region | 2.46 |
| Disallowed Region | 0.00 |
| Clashscore | 1.43 |
| PDB ID | 7RAQ |

[a]$R_{merge} = [\sum_h\sum_i | I_h - I_{hi} | /\sum_h\sum_i I_{hi}]$ where $I_h$ is the mean of $I_{hi}$ observations of reflection $h$. Numbers in parenthesis represent highest resolution shell. [b]$R_{factor}$ and [c]$R_{free} = \sum||F_{obs}| - |F_{calc}||/\sum|F_{obs}| \times 100$ for 95% of recorded data ($R_{factor}$) or 5% data ($R_{free}$). [d]Determined using MolProbity

The linear peptide in the CV3-25 structure contains one putative N-linked glycosylation site at Asn$_{1158}$. This glycan is not predicted to clash with CV3-25 binding to the peptide, but in the 6XR8 trimer with the extended stem helix, the glycan on one of the adjacent protomers would potentially clash with the heavy chain of the antibody (Fig. 4b). In the MD model, the glycan on the adjacent protomer shifts so that it is no longer clashing with the heavy chain (Fig. 4c). Additionally, the alignment of the model also suggests that the light chain of CV3-25 could potentially bind to the region just downstream of the stem helix at the start of HR2, $_{1168}$DISGINASVVN$_{1178}$ (Fig. 4d), a region that shows some sequence conservation amongst coronaviruses (Fig. 4e).

Superimposition of the CV3-25-peptide structure onto the post-fusion structure of SARS-CoV-2 spike (PDBid: 6XRA) reveals a different conformation of the CV3-25 epitope (Fig. 4f)[13]. In the post-fusion conformation, the stem helix in this epitope unwinds a full turn, relative to the CV3-25 bound peptide, with the remainder of the stem-helix elongating into a more linear structure. The overall RMSD between the CV3-25 bound peptide and this region in the post-fusion spike is 9.8 Å$^2$ over 15 Cα atoms and is therefore unlikely to be compatible with CV3-25 binding. CV3-25 inhibits spike mediated syncytia formation in vitro which depends on receptor engagement, cleavage of S2' to liberate the fusion peptide, and refolding of S2[71]. CV3-25 does not prevent the binding of an antibody to the ACE2 binding site (CV30, Fig. 2d), nor does it inhibit spike binding to cell surface expressed ACE2[50]. Collectively these observations suggest that CV3-25 is acting to disrupt fusion at a step following viral attachment, possibly by preventing the transition of the SARS-CoV-2 spike to the post-fusion state.

**Cross-reactivity of CV3-25 with the stem helix of other CoVs.** Several of the CV3-25 contact residues are conserved in beta-CoVs (Fig. 4e). We therefore evaluated the ability of CV3-25 to bind peptides derived from additional Beta-CoVs spanning the stem helix region by ELISA. CV3-25 Fab showed slightly weaker binding to peptides derived from MERS-CoV (20 nM), HCoV-HKU (31 nM), HCoV-OC43 (40 nM) than to a SARS-CoV-1/2/WIV1 peptide (5.2 nM) (Supplementary Table 2). CV3-25 IgG bound comparably peptides derived from SARS-CoV-1/2/WIV1, MERS-CoV, and HCoV-OC43 as measured by ELISA. CV3-25 IgG binding was slightly weaker to a peptide derived from HCoV-HKU1 (Fig. 5a). We did not observe any binding of CV3-25 to corresponding peptides from the alpha-CoVs HCoV-229E and HCoV-NL63 (Supplementary Fig. 4), consistent with a lack of CV3-25 binding to recombinant HCoV-229E and HCoV-NL63 spike proteins[50].

In contrast, the stem-helix directed mAbs B6 and CC40.8 showed differential binding to these peptides. B6 bound most strongly to the peptide from MERS-CoV, followed by HCoV-OC43, SARS-CoV-2 and HCoV-HKU1 (Fig. 5b), while CC40.8 exhibited the strongest binding to HCoV-HKU1, followed by MERS and HCoV-OC43 (Fig. 5c). We were unable to detect binding of CC40.8 to the SARS-CoV-2 peptide at the concentration tested here. To assess whether CV3-25 could bind to the linear epitope presented on these peptides in the context of a full-length spike protein, we expressed the membrane anchored, wildtype spike proteins from SARS-CoV-2, SARS-CoV-1, WIV1, HCoV-OC43, HCoV-HKU1, and MERS-CoV on the surface of 293 cells and stained them with fluorescently labeled CV3-25. We included B6, CC40.8, CV30 and AMMO1 mAbs for comparison. CV3-25 bound to SARS-CoV-2, SARS-CoV-1, WIV1, consistent with its ability to bind to the stem helix peptide from these spike proteins and neutralize the corresponding pseudoviruses (Fig. 5e–g). Despite binding to the stem helix peptide from MERS-CoV, HCoV-OC43 and HCoV-HKU1, and stabilized soluble versions of the corresponding spike ectodomains (Supplementary Fig. 5), CV3-25 did not recognize cell-surface expressed

unwinds a full turn, with the helix terminating Pro$_{1162}$ moving ~13 Å into the interior of the helix bundle (Fig. 4b). The unwinding and repositioning of the stem-helix by CV3-25 would create a clash between residues $_{1162}$PDVDL$_{1166}$ and the CDRH1 and CDRH3 of B6. There are additional potential clashes between the CDRH3 on CV3-25 and CDRH2 on B6. Collectively these provide a structural basis for the observed competition between B6 and CV3-25 binding to SARS-CoV-2 S2P, which was confirmed using a linear SARS-CoV-2 peptide (Supplementary Fig. 3). CV3-25 binding to the stem helix peptide also prevented subsequent binding of the CC40.8 mAb (Supplementary Fig. 3). CryoET analysis of the SARS-CoV-2 spike on the surface of intact virions shows an extended stalk region downstream of the stem helix[68,69]. However, all presently available cryoEM structures of the stabilized or membrane solubilized native spikes show poor resolution of the stem helix region and density for the downstream region including HR2 is missing[7,13,14]. A model of the full length spike ectodomain, including HR2 was determined using homology modelling and molecular dynamic (MD) simulation[70]. Aligning the CV3-25-stem helix structure to this model shows good agreement (Fig. 4c). In both the modeled and CV3-25-peptide structures, the stem helix ends at the glycosylated Asn$_{1158}$, and the extended C-terminal end of the CV3-25 bound peptide adopts a similar conformation to the MD model.

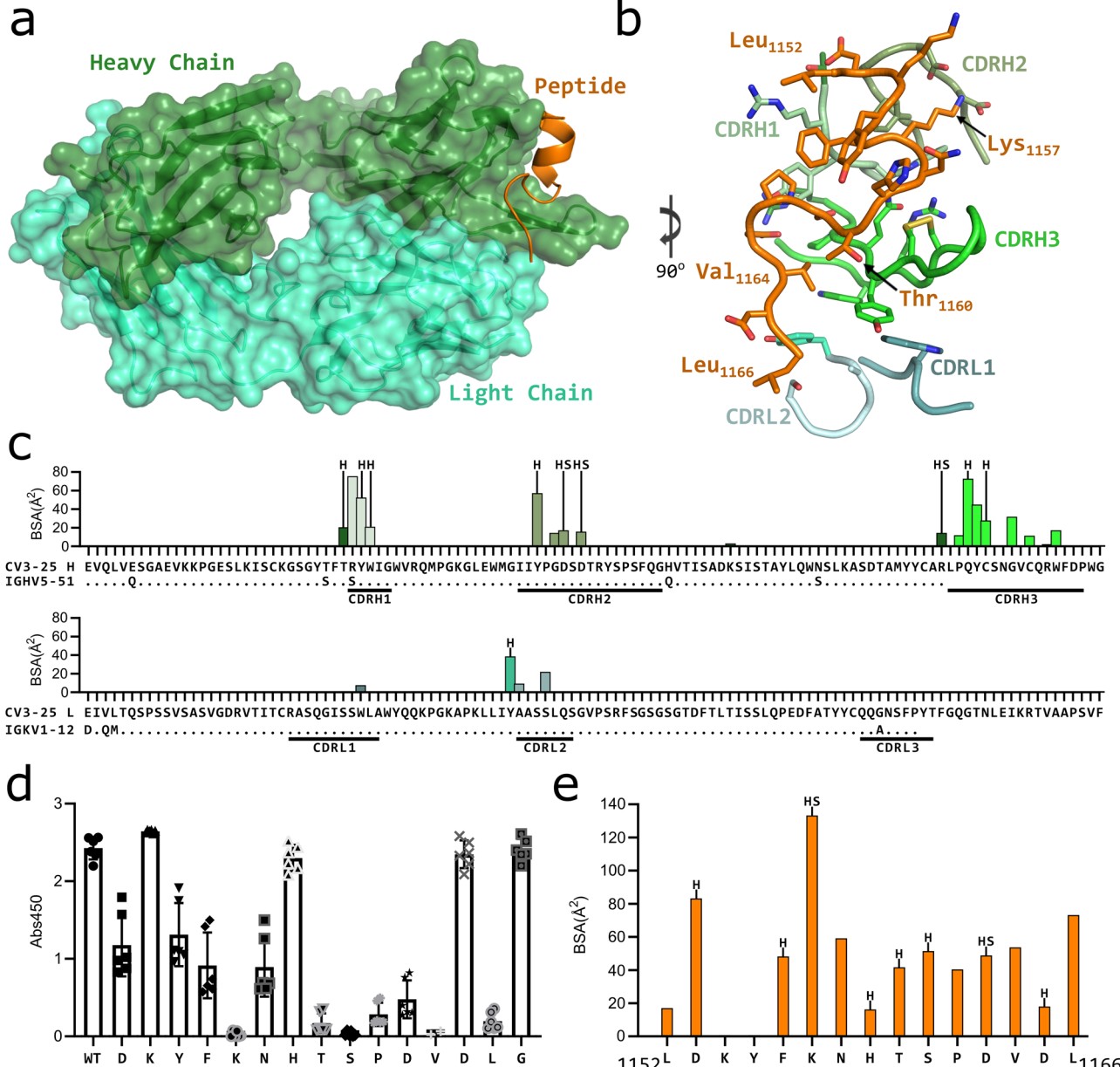

**Fig. 3 Structure of CV3-25 Fab bound to stem helix peptide. a** CV3-25-peptide shown in ribbon structure with mAb surface representation shown in transparency. CV3-25 heavy chain is shown in green and light chain in cyan. The peptide is shown in orange. **b** Details of the interactions between the Fab and the peptide. Complementary determining regions (CDRs) are shown in ribbon representation and labeled and colored as shown. Interacting sidechains of the Fab are shown and some of the important residues in the peptide are labeled. **c** Plots of buried surface area (BSA) of each Fab residue interacting with the peptide and a sequence alignment with the corresponding V-gene. CDRs are labelled and color coded to match the structure shown in (**b**). Residues engaged in a hydrogen bond or salt bridge are marked with an "H" or "S", respectively. **d** Alanine scanning plot of the stem helix region that CV3-25 binds. CV3-25 binding to linear peptides corresponding to amino acids 1153-1167 of the SARS-CoV-2 spike, where each amino acid was substituted by alanine was measured by ELISA. The absorbance at 450 nm resulting from the addition of 1.25 µg of CV3-25 is shown. Each dot represents a technical replicate from three independent experiments conducted in duplicate. Bars represent the mean and error bars represent the standard deviation. Full titrations are shown in Fig. S1. **e** Plot of the BSA of each stem helix peptide residue.

spikes (Fig. 5h–j). In line with the lack of binding, CV3-25 failed to neutralize a MERS-CoV pseudovirus or authentic HCoV-OC43 (Fig. 5k, m). Similarly, a monovalent Fab was unable to neutralize HCoV-OC43, indicating that the lack of neutralization was not due to steric shielding of the epitope from full length IgG (Fig. 5l). CV3-25 was also unable to neutralize authentic HCoV-NL63, an alpha-CoV (Supplementary Fig. 6). We conclude that although the CV3-25 epitope is present, it is not equally accessible in the native conformation of the spike protein among the various beta-CoVs examined here.

**Somatic mutation leads to stronger cross-reactive binding by CV3-25**. To assess the role of somatic mutation in CV3-25 cross-reactivity we measured the binding of iGL-CV3-25 to the same linear peptides from SARS-CoV-2, MERS-CoV, HCoV-OC43 and HCoV-HKU1 by ELISA (Fig. 5d). Although the binding to the peptide from SARS-CoV-1/2 and WIV1 was comparable and strong, the binding was severely reduced to MERS and OC43, and to a lesser extent to HKU1. Thus, somatic mutations acquired by CV3-25 lead to broad CoV-peptide reactivity.

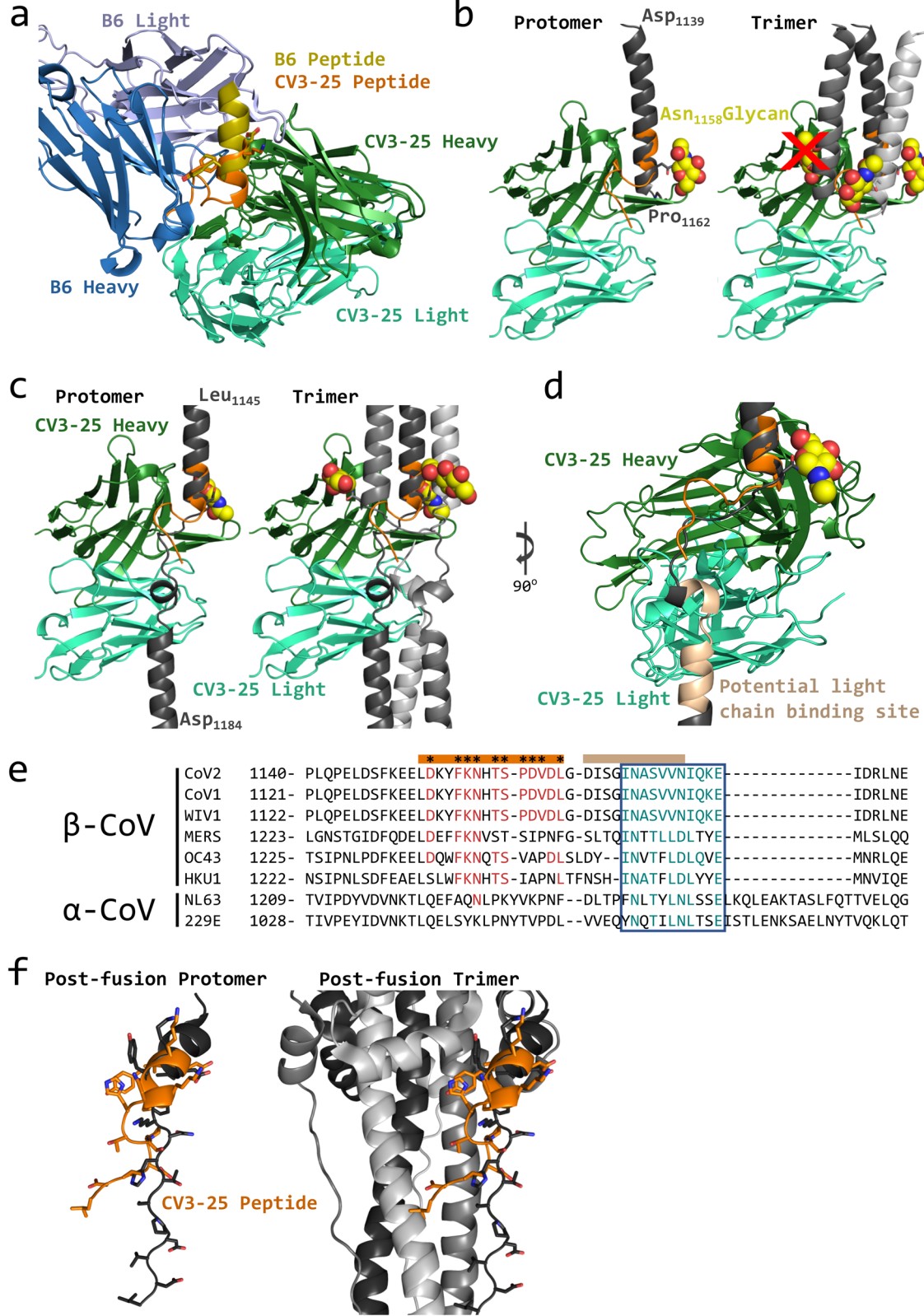

## Discussion

The devastating loss of life, economic and social impacts of the SARS-CoV-2 pandemic underscores the need to prevent future CoV outbreaks. The fact that SARS-CoV-2 is the third highly pathogenic CoV to cause significant loss of human life in the past two decades suggests that future CoV outbreaks are plausible if not inevitable. Since neutralizing antibodies are likely important for protection against CoV infection, the isolation and characterization of mAbs targeting conserved neutralizing epitopes present across CoV variants and strains can inform the design of pan-CoV vaccines that can prevent or blunt future outbreaks.

So far, six neutralizing mAbs targeting the stem-helix region, elicited by immunization in mice, or humanized mice (B6, 1.6C7, 28D9 and IgG22), or isolated from SARS-CoV-2 infected humans

**Fig. 4 Structural basis for broad CoV recognition by CV3-25. a** Structural alignment of stem helix peptides to CV3-25 Fab and B6 Fab (PDBid: 7M53) shown as ribbon diagram. B6 heavy chain is shown in dark blue and light chain in light blue. CV3-25 heavy chain is shown in green and light chain in cyan. The B6-bound peptide is shown in yellow and the CV3-25-bound peptide in orange. **b** Left, structural alignment of the CV3-25 structure and the stem helix structure (PDBid: 6XR8) shown in cartoon representation. The 6XR8 stem helix is shown in dark gray and the CV3-25-bound peptide is shown in orange. Asn$_{1158}$ glycan is shown in yellow sphere representation. Left, the alignment with a single protomer, right, an alignment with the trimer with the location of the glycan clash marked by a red X. **c** Structural alignment of CV3-25 to MD simulation model of the stem helix and HR2 region of the spike protein[70]. Asp$_{1158}$ glycan is shown in yellow sphere representation. Left, an alignment with a single protomer, right an alignment with the trimer. **d** Potential interaction area of the N-terminal end of HR2 and the light chain of CV3-25 is shown in tan. **e** Sequence alignment of the stem helix region of several CoV spike proteins. The peptide bound to CV3-25 is marked by the orange bar and crucial interacting residues are marked by *. The residues conserved in the binding site are shown in red. The region that could interact with the light chain is shown with the tan bar and the conserved region is highlighted by the blue box with similar residues shown in teal. **f** Structural alignment of the CV3-25 bound peptide (orange) to the post-fusion S (black, PDBid: 6XRA). The peptide was aligned to $_{1152}$LDKY$_{1155}$ in the spike. Left, the alignment to the protomer. The sidechains of the residues are shown in stick representation, right, the region is shown in context of the trimer.

(CC40.8, S2P6, CV3-25), have been described[50,57–59,66,72]. All show varying degrees of cross-reactivity and cross-neutralizing activity against CoVs. Among these mAbs, four S2P6, CC40.8, IgG22 and B6 have previously been structurally characterized. All form a hydrophobic groove that cradles the hydrophobic face of the amphipathic SARS-CoV-2 stem helix (residues AA 1147-1156 for B6, residues 1142-1159 for CC40.8, 1147-1156 for IgG22, and residues 1146-1159 for S2P6). Alanine scanning of a stem helix peptide from MERS S suggests that the 1.6C7 and 28D9 mAbs which were isolated from humanized mice immunized sequentially with recombinant S from OC43, SARS and MERS, bind to a similar epitope region[64]. In contrast to the other stem helix mAbs, CV3-25 binds to a distinct S2 epitope C terminal and on the opposite face of the amphipathic stem helix thus defining an additional site of vulnerability on pathogenic beta-CoVs.

Very little high-resolution structural information is available about the conformation of the stem helix, and region C-terminal to the stem-helix. As such, we can only speculate as to why CV3-25 binds well to recombinant spike ectodomains and linear peptides from the MERS-CoV, HCoV-OC43 and HCoV-HKU1 beta-CoVs, but fails to bind full-length cell-surface expressed spikes or neutralize the corresponding viruses/pseudoviruses. CryoEM structures of stabilized NL63, MERS-CoV, SARS-CoV, and SARS-CoV-2 spike proteins only resolve the N-terminus of the stem-helix region[7,14,73–77]. Moreover, complexes of mAbs B6, S26P, IgG22 and CC40.8 with stabilized spikes from MERS, SARS-CoV-2 and HKU1 are also poorly resolved by EM suggesting that this region undergoes significant conformational heterogeneity on recombinant proteins[57,58,65,66]. Subtomogram averaging of virion-anchored SARS-CoV-2 spikes show evidence of flexible hinges in the stalk region[68,69]. The stalk has been likened to a leg with a hip, knee and ankle joint where the stem helix region corresponds to the upper leg[68]. CV3-25 would bind to the dynamic "knee" region and may only recognize one of many conformations adopted by the SARS-CoV-2 spike. If so, it's possible that a similar conformation is not sampled by membrane-anchored MERS, HKU1 or OC43 spikes. Alternatively, the CV3-25 epitope could be less exposed in the context of the MERS, OC43 and HKU1 membrane-anchored spikes as compared to their corresponding stabilized ectodomains or the membrane-anchored sarbecovirus spikes. We note however, that the smaller size Fab domain of CV3-25 was unable to neutralize the OC43 virus.

It has been proposed that the B-cell lineages that gave rise to the stem helix mAbs CC40.8 and S26P were initiated by previous OC43 and HKU1 infection, respectively[58,65]. Despite binding to linear peptides from HKU1 and to a lesser extent OC43, the absence of CV3-25 reactivity with membrane-anchored spikes from endemic HCoVs suggests that the CV3-25 progenitor B cell was activated by the SARS-CoV-2 virus rather than a prior HCoV-infection.

The neutralizing potency of CV3-25 is not affected by mutations found in SARS-CoV-2 variants of concern (Alpha, Beta, Delta, Gamma and Omicron) which harbor mutations that escape from many anti-NTD and anti-RBD antibodies. In line with this the linear peptide bound by CV3-25 is invariable in the variants of concern, and in the WHO-defined variants of interest (Eta, Iota, Kappa Lambda and Mu). Moreover, the CV3-25 epitope is strictly conserved among SARS-CoV-1, SARS-CoV-2 and WIV1 as well as several other sarbecovirus isolates from bats and pangolins (Supplementary Fig. 7).

The observations that CV3-25 shows broad sarbecovirus neutralization and displays anti-viral activity in K18-hACE2 mice, particularly when Fc receptors are engaged[71], indicates that the CV3-25 epitope is highly relevant to the development of a pan-sarbecovirus vaccine. The fact that CV3-25 binds a linear epitope indicates that it may be possible to design small scaffold based, or subunit vaccines that present the CV3-25 epitope while avoiding eliciting an immunodominant response to non-neutralizing epitopes on S2 and elsewhere on the spike. The observation that CV3-25 competes for binding with B6 and CC40.8 despite binding to discrete linear epitopes, indicates that multiple scaffold design strategies may need to be employed to target these two conserved sites of CoV vulnerability in the stem-helix region in order to provide broad neutralizing coverage against diverse CoV. Similarly, a therapeutic combination of non-competing stem-helix mAbs may provide broad neutralizing coverage against emergent pathogenic CoVs since CV3-25 neutralizes diverse sarbecoviruses, and B6 neutralizes multiple merbecoviruses and OC43, a member of the embecovirus subgenus[57].

## Methods

**Cell lines**. All cell lines were incubated at 37 °C in the presence of 5% CO2. 293-6E (human female, RRID:CVCL_HF20) and 293 T cells (human female, RRID:CVCL_0063) cells were maintained in Freestyle 293 media with gentle shaking. HEK-293T-hACE2 (human female, BEI Resources Cat# NR-52511) were maintained in DMEM containing 10% FBS, 2 mM L-glutamine, 100 U/ml penicillin, and 100 μg/ml streptomycin (cDMEM). HCT-8 [HRT-18] cells (human male, ATCC CCL-244) were maintained in RPMI containing 10% horse serum, 2 mM L-glutamine, 100 U/ml penicillin, and 100 μg/ml streptomycin. LLC-MK2 cells (*Macaca mulatta*, ATCC CCL-7) were maintained in cDMEM. Huh7 cells (human male, a gift from Dr. Ram Savan, Department of Immunology University of Washington) were maintained in cDMEM. None of the cell lines used were authenticated or tested for mycoplasma contamination.

**Recombinant CoV proteins and mAbs**. Two stabilized versions of the recombinant SARS-CoV-2 spike protein (SARS CoV-2 6P-D614G[67] and S6P[78]) and the SARS-CoV-2 RBD were produced in 293E or 293 F cells and purified as previously described[50,67,79]. Plasmids encoding the stabilized versions of HCoV-OC43 (619-M66-303: CMV51p > HCoV-OC43 S-2P-T4f-3C-His8-Strep2x2, Addgene plasmid # 166015) and HCoV-HKU1 (R619-M89-303: CMV51p > HCoV-HKU1 S-2P-T4f-3C-His8-Strep2x2, Addgene plasmid # 166014) spike proteins were gifts from Domonic Esposito. The proteins were expressed in 293E cells and purified using Ni-NTA affinity resin followed by size exclusion chromatography on a superose 6 column as described in[80]. Recombinant CV3-25, CV30, B6 and AMMO1 were

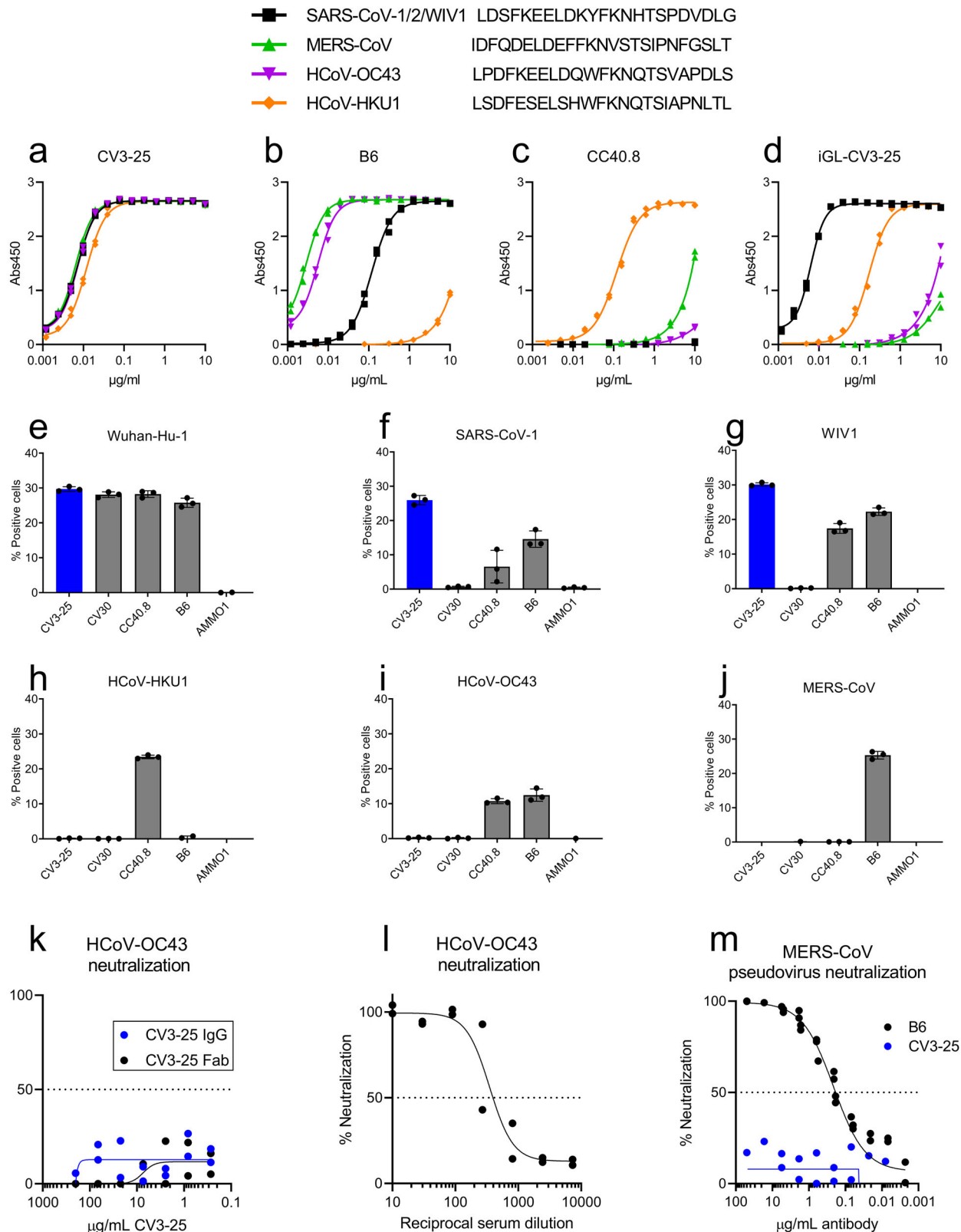

expressed in 293 cells and purified using protein A resin as previously described[50,57,62].

**Generation of plasmids expressing SARS-CoV-2 spike variants and MERS-CoV**. To generate a plasmid encoding the SARS-CoV-2 spike P.1 variant (pHDM-SARS-CoV-2-Spike-P.1) primers were designed that anneal 5′ of the L18 codon and just 3′ of the V1176F codon on the pHDM-SARS-CoV-2 Spike Wuhan-Hu-1 plasmid (BEI Resources Cat# NR-52514) and used to amplify cDNA corresponding

to the N and C termini of the spike protein and the plasmid backbone using Platinum SuperFi II DNA Polymerase (Thermofisher Cat# 12368010) according to the manufacturer's instructions. cDNA encoding the rest of the spike protein including the Δ242-243 deletion and the L18F, T20N, P26S, D138Y, R190S, K417T, E484K, N501Y, D614G, H655Y, T1027I, and V1176F mutations was synthesized as two gBlocks (Integrated DNA technologies). The first had 30nt of homology with the PCR amplified vector backbone at the 5′ end. The second included 30nt of homology with the 3′ end of the first block at the 5′ end and 30nt of homology with

**Fig. 5 CV3-25 binds to stem helix peptides from diverse betacoronaviruses but only to cell surface-expressed sarbecovirus spike proteins.** Binding of CV3-25 (**a**), B6 (**b**), CC40.8 (**c**) or the inferred germline version of CV3-25 (**d**), to linear peptides from SARS-CoV-1/2/WIV1, MERS-CoV, HCoV-OC43, and HCoV-HKU1 was measured by ELISA. Each dot represents a technical replicate measured in duplicate in (**a–d**). Spike proteins from SARS-CoV-2 Wuhan-Hu-1 (**e**), SARS-CoV-1 Urbani (**f**), WIV1 (**g**), HCoV-HKU1 (**h**) and HCoV-OC43 (**i**) and MERS-CoV (**j**) spike proteins were expressed on the surface of 293 cells, and then stained with the indicated fluorescently labeled mAbs and then analyzed by flow cytometry. The percentage of cells that stained positive with the mAbs is indicated on the y axis. One representative example of two independent experiments with three technical replicates is shown. Neutralization of authentic OC43 by CV3-25 IgG or Fab (**k**), or human sera (**l**). **m** Neutralization of MERS-CoV pseudovirus by the indicated mAbs. Each dot represents a technical replicate from one or two experiments conducted in duplicate in (**a–m**). Bars represent the mean and error bars represent the standard deviation in (**e–j**).

the PCR amplified vector backbone at the 3′ end. The gBlocks and PCR product were ligated together using InFusion HD cloning Plus (TakaraBio Cat#638920). To generate a plasmid encoding the SARS-CoV-2 spike B.1.1. variant (pHDM-SARS-CoV-2-Spike-B.1.1.7) primers were designed that anneal 5′ of the H69 codon and just 3′ of the D1118 codon on the pHDM-SARS-CoV-2 Spike Wuhan-Hu-1 plasmid (BEI Resources Cat# NR-52514) and used to amplify cDNA corresponding to the N and C termini of the spike protein and the plasmid backbone using Platinum SuperFi II DNA Polymerase (Thermofisher Cat# 12368010) according to the manufacturer's instructions. cDNA encoding the rest of the spike protein including the H69-V70 and Y144 deletions, N501Y, A570D, D614G, P681H, T716I, S982A and D1118H mutations. A plasmid encoding the SARS-CoV-2 Spike B.1.617.2 (pCMV3-SARS-CoV-2-Spike-B.1.617.2) was purchased from Sinobiological (Cat# VG40804-UT).

To generate a plasmid encoding the MERS-CoV-2 spike (pHDM-MERS-CoV-Spike) codon-optimized cDNA corresponding to the MERS-CoV S protein (Riyadh_14_2013, GenBank: AHI48572.1) flanked on the 5′ end by 30 nt of homology upstream and including the EcoRI site and flanked on the 3′ end by 30 nt of homology downstream of and including the HindIII site on the pHDM-SARS-CoV-2 Spike Wuhan-Hu-1 plasmid was synthesized by Twist Biosciences. The synthesized DNA was cloned into the pHDM-SARS-CoV-2 Spike Wuhan-Hu-1 plasmid that was cut with EcoRI and HindIII and gel purified to remove the SARS-CoV-2 Spike cDNA using InFusion HD cloning Plus. The sequences of the cDNA of all the spike expression constructs were verified by Sanger sequencing (Genewiz Inc.).

**Peptides.** Peptides were synthesized by Genscript or A&A Labs with, or without a biotin molecule conjugated to the amino-terminus via aminohexanoic acid.

**Pseudovirus neutralization assay.** HIV-1 derived viral particles were pseudo-typed with full length wildtype S from Wuhan Hu1, B.1.351, B.1.1.7, P.1, WIV1, or MERS-CoV using a previously described reporter system[81]. Briefly, plasmids expressing the HIV-1 Gag and pol (pHDM540 Hgpm2, BEI Resources Cat# NR-52517), HIV-1Rev (pRC-CMV-rev1b, BEI Resources Cat# NR-52519), HIV-1 Tat (pHDM-tat1b, BEI resources NR-52518), SARS-CoV-2 spike (pHDM-SARS-CoV-2 Spike Wuhan-Hu-1, pHDM-SARS-CoV-2-Spike-B.1.1.7, SARS-CoV-2 Spike-P.1, pHDM-SARS-CoV-2 Spike-B.1.351[30], pCMV3-SARS-CoV-2-Spike-B.1.617.2, pTWist-WIV1-CoV (a gift from Alejandro Balazs (Addgene plasmid # 164438; http://n2t.net/addgene:164438; RRID:Addgene_164438), or pHDM-MERS-CoV Spike and a luciferase/GFP reporter (pHAGE-CMV-Luc2-IRES542 ZsGreen-W, BEI Resources Cat# NR-52516) were co-transfected into 293 T cells at a 1:1:1:1.6:4.6 ratio using 293 Free transfection reagent according to the manu-facturer's instructions. Pseudoviruses lacking a spike protein were also produced as a control for specific viral entry. Pseudoviron production was carried out at 32 °C for 72 h after which the culture supernatant was harvested, clarified by cen-trifugation and frozen at −80 °C.

293 cells stably expressing human HEK-293T-hACE2, for SARS-CoV-2 pseudoviruses, or Huh-7 cells for MERS-CoV pseudoviruses were seeded at a density of $4 \times 10^3$ cells/well in a 100 µL volume in 96-well flat bottom black-walled, clear bottom tissue culture plates (Greiner CELLSTAR Cat# 655090). The next day, mAbs were serially diluted in 70 µL of cDMEM in 96-well round bottom master plates in duplicate wells. 30 µL of serially diluted mAbs from the master plate were replica plated into 96-well round bottom plates. An equal volume of viral supernatant was added to 96-well round bottom plates containing identical serial dilutions from the same master plate, and incubated for 60 min at 37 °C. Meanwhile 50 µL of cDMEM containing 6 µg/mL polybrene was added to each well of 293T-ACE2 or Huh-7 target cells and incubated for 30 min. The media was aspirated from target cells and 100 µL of the virus-antibody mixture was added. The plates were incubated at 37 °C for 72 h. The supernatant was aspirated and replaced with 100 µL of Steady-Glo luciferase reagent (Promega Cat# E2510) and read on a Fluoroskan Ascent Fluorimeter. Control wells containing virus but no antibody (cells + virus) and no virus or antibody (cells only) were included on each plate.

Percent neutralization for each well was calculated as the RLU of the average of the cells + virus wells, minus test wells (cells + mAb + virus), and dividing this result difference by the average RLU between virus control (cells + virus) and

average RLU between wells containing cells alone, multiplied by 100. The antibody concentration that neutralized 50% or 80% of infectivity (IC50 and IC80 for mAbs) was interpolated from the neutralization curves determined using the log(inhibitor) vs. response–Variable slope (four parameters) fit using automatic outlier detection in GraphPad Prism Software.

**Biolayer interferometry (BLI).** BLI experiments were performed on an Octet Red instrument at 30 °C with shaking at 500–1000 rpm.

Kinetics analysis:

Streptavidin (SA) biosensors (Fortebio) were immersed Kinetics Buffer (KB: 1X PBS, 0.01% Tween 20, 0.01% BSA, and 0.005% NaN3, pH 7.4) containing 10 µg/ml of biotinylated peptides or biotinylated S6P for 150 s by immersion in KB for 60 s to achieve a baseline reading. Probes were then immersed in KB containing serially diluted CV3-25 Fab for a 300 s association phase, followed by a 300 s dissociation phase in KB. The background signal from each analyte-containing well was measured using empty reference sensors and subtracted from the signal obtained with each corresponding mAb loaded sensor. Kinetic analyses were performed at least twice with an independently prepared analyte dilution series. Curve fitting was performed using a 1:1 binding model and the ForteBio data analysis software. Mean kon, koff values were determined by averaging all binding curves that matched the theoretical fit with an R2 value of ≥0.98.

Binding competition assays: SA biosensors were immersed in KB containing 10 µg/ml of biotinylated peptides or S6P for 300 s, followed by a 20 s baseline in KB buffer. Probes were then immersed in KB containing 20 µg/ml CV3-25, AMMO1, B6 or CC40.8 for a 300 s association phase, followed by a 20 s baseline in KB buffer and then immersed into KB containing 20 µg/ml CV3-25, AMMO1, B6 or CC40.8 for a 300 s association phase.

**ELISA.** MaxiSorp microtiter plates (Thermo Scientific Cat#464718) were coated with 300 ng/well of streptavidin (New England Biolabs Catalog #: N7021S) over-night at room temperature. Plates were washed 4X with PBS with 0.02% Tween-20 (wash buffer), then incubated with 60 µL/well of 3% BSA and 0.02% Tween-20 in PBS (blocking buffer) for 1 hr at 37 °C. After washing 4X with wash buffer, 380 ng/well of biotinylated peptides diluted in blocking buffer were incubated for 1 hr at 37 °C. Plates were washed 4X in wash buffer and then mAbs were serially diluted in blocking buffer, added to the plate and incubated for 1 hr at 37 °C. Plates were washed 4X in wash buffer and the secondary antibody Goat anti-Human Ig-HRP (Southern Biotech, Cat# 2010-05), was added and incubated at 37 °C for 1 hr. Plates were washed 4X wash buffer, and then 30 µL/well of SureBlue Reserve TMB Per-oxidase Substrate (Seracare KPL, Cat# 5120-0080) was added and incubated for 3 min followed by addition of 30 µL of 1 N H2SO4 to stop the reaction. The optical density at 450 nm was measured using a SpectraMax i3x plate reader (Molecular Devices). All wash steps were performed using a BioTek 405/TSMicroplate Washer.

**Negative stain electron microscopy.** Negative stain electron microscopy was performed as previously described[67]. In brief, IgG was added to stabilized S protein (SARS-CoV-2-6P-D614G) at a 3-fold molar excess and incubated at room tem-perature for 30 min. The complex was then diluted to approximately 0.03 mg/mL in 1x TBS pH 7.4, added onto a carbon coated 400 square copper mesh grid, and immediately stained with 2% Nano-W (Nanoprobes) for 7 s and then again for 14 s. Imaging was performed with the Leginon automated data collection software[82] on a 120 keV FEI Tecnai Spirit electron microscope using a Thermo Fisher Eagle 4k x 4k CCD camera at 52,000x magnification, −1.50 µm defocus and 2.06 Å pixel size. Particles were picked using DoGPicker[83] via Appion[84] and processed in RELION 3.0[85]. An initial model generated from a published SARS-CoV-2 S protein structure (PDB: 6VYB[7]) was used during data processing. Using 2D classes as references, particles with a visible S2 stem but no antibody density were selected and refined against the initial model described above. Next, particles with clear antibody density were selected and refined (with C3 symmetry) against the elongated stem 3D map. These refined particles were subjected to C3 symmetry expansion. A 40 Å spherical mask was placed at the expected Fab region near the S2 stem and focused classifications were performed without alignment. 2 classes had density resembling Fab and were selected. Duplicate particles were removed and the remaining stack was refined without symmetry. To better align the Fab density and prevent misalignment due to the remaining regions of IgG, a spherical

mask of diameter 320 Å or a tighter mask of trimer and Fab only were used in a final stage of refinement. Segmentation for illustrations was performed using Segger[86] in USCF Chimera[87].

**Production of CV3-25 Fab**. Purified recombinant CV3-25 IgG was mixed with LysC (NEB) at a ratio of 1 µg LysC per 10 mg of IgG and incubated at 37 °C for 18 h with nutation. The cleaved product was incubated with 1 mL of Protein A resin (GoldBio) per 10 mg of initial IgG and incubated at room temp for 1 hr to bind any uncleaved IgG and digested Fc. The purified Fab was further purifed by SEC using a HiLoad 16/600 Superdex 200 pg column.

**Crystal screening and structure determination**. CV3-25 Fab was incubated with a 1.5 molar excess of the synthetic stem helix peptide spanning residues 1149–1167 (Genscript). Initial crystal screening was performed by sitting-drop vapor-diffusion in the MCSG Crystallization Suite (Anatrace) using a NT8 drop setter (Formulatrix). Poorly diffracting crystals grew in MCSG-3 well B1 and were optimized using the Additive Screen (Hampton Scientific). Diffracting crystals were obtained in a mother liquor (ML) containing 0.1 M Na Acetate:HCl, pH 4.5, 2.0 M $(NH_4)_2SO_4$, 0.1 M Strontium Chloride. The crystals were cryoprotected by soaking in ML supplemented with 26% glycerol. Diffraction data were collected at Advanced Light Source beamline 5.0.2 at 12286 keV. The data set was processed using XDS[88] and data reduction was performed using AIMLESS in CCP4[89] to a resolution of 1.74 Å. Initial phases were solved by molecular replacement using Phaser[90] in Phenix[91,92] with a search model of Fab 4AB007 (PDBid: 5MVZ) divided into Fv and Fc portions. Model building was completed using COOT[93] and refinement was performed in Phenix with the final refinement run through the PDB_REDO server[94]. The data collection and refinement statistics are summarized in Table 1. Structural figures were made in Pymol (Schrodinger, LLC).

**Cell surface SARS-CoV-2 S binding assay**. cDNA corresponding to AA 15-1336 of HCoV-OC43 was PCR amplified from pCAGGS-Flag-HCoV-OC43 Spike (a kind gift from Dr. Marceline Côté, University of Ottawa) and cloned into the pTT3 vector using InFusion cloning (Clontech). A Kozak consensus sequence and the TPA leader sequence (MDAMKRGLCCVLLLCGAVFVSPSAS) was added to the 5′ end of the cDNA during PCR amplification. cDNA for the HKU1 spike was PCR amplified from pCMV-HCoV-HKU1 (SinoBiological Cat# VG40606-UT) and subcloned into pTT3.

pTT3-SARS-CoV-2-S[79], pHDM-MERS-CoV-Spike, pTWist-WIV1-CoV, pHDM-MERS-CoV-1 Spike, pTT3-HKU1 or pTT3-OC43 Spike were transfected into suspension-adapted 293 T cells using 293 Free transfection reagent (EMD Millipore Cat# 72181) or PEI transfection reagent (PolySciences Inc. Cat# 23966) according to the manufacturer's instructions. Transfected cells were incubated for 24 h at 37 °C with shaking. Meanwhile, 1 µg of each mAb was added to individual wells of a 96 well plate in 50 µl of FACS buffer (PBS + 2% FBS + 1 mM EDTA).

Spike-transfected or mock-transfected 293 T cells were resuspended at $4 \times 10^6$ cells/ml in FACS buffer and 50 µl was added to each well of the 96 well plate. mAb-cell mixture was incubated for 30 min on ice. The plates were then washed once with 200 µl of FACS buffer and stained with of PE-conjugated, AffiniPure Fab fragment goat anti-human IgG (Jackson Immunoresearch Cat# 109-117-008) at a 1:100 dilution and live/dead green fluorescent reactive dye (Thermo Fisher Cat# L34970) at a 1:1000 dilution in 50 µl/well of 1X PBS. The staining reaction was incubated for 20 min in the dark on ice. The plates were then washed once with 200 µl of FACS buffer and fixed with 50 µl of 10% formalin. The plates were centrifuged, and the formalin was removed and replaced with 250 µl of FACS buffer. The % of live PE + cells was measured on a Guava easyCyte 5HT Flow Cytometer (Luminex). For each mAb, the % of PE + mock transfected cells was subtracted from the % of PE + of spike transfected cells. An example of the gating strategy is shown in Supplementary Fig. 7.

**OC43 live virus neutralization assay**. HCT-8 [HRT-18] cells (ATCC CCL-244™) were seeded at 20,000 cells/well in 96-well plates and cultivated in RPMI containing 10% horse serum and penicillin-streptomycin at 37 °C for 2 days until reaching near confluency. Fifty-fold of the fifty percent tissue culture infection doses (TCID$_{50}$) of OC43 (Zeptometrix Cat#0810024CF) per well was used. Serially diluted serum or mAb was mixed with virus in serum-free RPMI and incubated for 1 h at 33 °C on a shaker at 150 rpm. Then, the virus:antibody mixture was transferred onto HCT-8 cells and the plate was incubated at 33 °C in a CO$_2$ incubator. At day 5, cells were fixed with −20 °C-cold, 70% MeOH for 15 min. Plate was rinsed with PBS (Gibco) and blocked with PBS containing 2.5% Blotting grade blocker (Bio-Rad) and 0.05% Tween-20 (Sigma) for 1 hr at 37 °C. After washing one time with PBS-T (PBS, 0.05% Tween-20), plate was incubated with rabbit anti-nucleocapsid antibody (Sino Biological, Cat# 40643-T62) for 1 h at room temperature on a plate shaker at 800 rpm. After that, plate was washed three times with PBS-T and incubated with goat anti-rabbit IgG-HRP (Jackson ImmunoResearch, Cat# 111-035-144) for 1 h at room temperature on a plate shaker at 800 rpm. After washing three times with PBS-T, the assay was developed by addition of 1-Step Ultra TMB-ELISA solution (Thermo Scientific Cat# 34028) and reaction was stopped with 2 N sulfuric acid (Fisher Scientific). Optical density at 450 and 620 nm was captured with SpectraMax M2 (Molecular Devices). Neutralization was defined as the antibody concertation that reduced OD relative to virus control wells (cells + virus only) after subtraction of background OD in cells-only control wells.

**NL63 live virus neutralization assay**. LLC-MK2 cells were seeded at 3000 cells/well in 96-well plate and cultivated in DMEM containing 10% FBS and penicillin-streptomycin at 37 °C for 2 days until reaching near confluency. Fifty-fold of the fifty percent tissue culture infection dose (TCID$_{50}$) of NL63 (BEI resources Cat# NR-470) per well was used. CV3-25 was added to a final concentration of 400 µg/ml and mixed with virus in serum-free RPMI and incubated for 1 h at 33 °C on a shaker at 150 rpm. Wells containing PBS and no virus were included as controls. After the virus and/or virus:antibody mixture was transferred onto LLC-MK2 cells and the plate was incubated at 33 °C in CO$_2$ incubator. Eight days later the cells were visually inspected for evidence of cytopathic effects under a light microscope.

**Statistics and reproducibility**. The number of replicates for each experiment are indicated in the figure legends. Individual data points and mean ± standard error of mean (SEM) are shown for bar graphs. Log(inhibitor) vs. response–Variable slope (four parameters) curves for neutralization assays, and ELISAs were performed in Graph Pad Prism v9.0.

**Reporting summary**. Further information on research design is available in the Nature Research Reporting Summary linked to this article.

## Data availability

All data are available in the manuscript or the Supplementary Material. The source data for the graphs and charts are available as Supplementary Data files 1, 2 and 3, and any remaining information can be obtained from the corresponding author upon reasonable request. The CV3-25/peptide structure has been deposited in the PDB (7RAQ). The negative stain EM map of CV3-25 IgG in complex with SARS-CoV-2 6P-D614G S protein has been deposited to the Electron Microscopy Data Bank under accession code EMD-25498. All reagents generated in this study are available upon request through Material Transfer Agreements. pTT3-derived plasmids and 293-6E cells require a license from the National Research Council (Canada).

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

## Acknowledgements

This work was supported by generous donations to Fred Hutch COVID-19 Research Fund (L.S.), Fast Grants (part of Emergent Ventures at George Mason University) and a COVID pilot award from the Fred Hutch (J.B.), the M.J. Murdock Charitable Trust (A.T.M), and The Bill and Melinda Gates Foundation OPP1170236 and INV-004923 (A.B.W.). X-ray diffraction data was collected at the Berkeley Center for Structural Biology beamline 5.0.2 at the Advanced Light Source (ALS), which is supported in part by the Howard Hughes Medical Institute. ALS is a Department of Energy Office of Science User Facility under Contract No. DE-AC02-05CH11231. We thank the James B. Pendleton Charitable Trust for its generous support of Formulatrix robotic instruments. We thank Stephen C. DeRosa and Kristen W Cohen for providing some of the peptides used in this study.

## Author contributions

Conceptualization: A.T.M., M.P., and L.S.; Investigation: N.K.H., L.J.H., I.S., M.F.J, A.J.M, Y-H.W., P.Z., J.B., A.M.H., A.M.J. Writing - Original Draft: N.K.H., A.T.M, M.P. and L.S.; Writing - Review & Editing: N.K.H., L.J.H., I.S., M.F.J, A.J.M., Y-H.W., P.Z., J.B., A.M.H., A.M.J, D.R.B., R.A., G.O. A.B.W., L.S., M.P., A.T.M.; Funding Acquisition: L.S. A.B.W. and A.T.M.

## Competing interests

L.S., M.P., and A.T.M. are inventors on a provisional patent application No. 63/131599 filed by the Fred Hutchinson Cancer Research Center on the CV3-25 monoclonal antibody. A.T.M. is an inventor on a provisional patent application No. 63/108,554 filed by the University of Washington on the B6 monoclonal antibody. All other authors declare no competing interests.
