## [Peer Review File · Communications Biology]

REVIEWERS' COMMENTS:

Reviewer #1 (Remarks to the Author):

Most of the major issues I raised have now been addressed and I believed the manuscript has now been improved and is almost ready for publication.

The only important issue left is that my comment 3 has been ignored and the old review replaced by a new one that has appeared in the meantime. This is unacceptable -- since the authors want to make a major point about fusion they should really cite the original research in lines 77-83 and in other places throughout the manuscript where they refer to all the spike activities I mentioned rather than just reference a review.

Two more minor comments:

1. line 526 indicates pixel size for a CCD camera with a tenth of picometer precision which seems rather inflated
2. The schematic of the EM data processing (Supp Fig. 1) is lacking some statistical/numerical data processing information: how many micrographs were analysed, how many particles went into each classification etc. etc.

Our response to the reviewer comments are in bold.

Reviewer #1 (Remarks to the Author):

The only important issue left is that my comment 3 has been ignored and the old review replaced by a new one that has appeared in the meantime. This is unacceptable -- since the authors want to make a major point about fusion they should really cite the original research in lines 77-83 and in other places throughout the manuscript where they refer to all the spike activities I mentioned rather than just reference a review.

We have now provided primary references for the SARS-CoV-2 spike activities including furin and TMPRSS cleavage, ACE2-binding and references for the pre-and post-fusion structures (Refs. 6-12) and kept the two reviews we cited in the previous submission; one on CoV fusion in general (Ref. 5) and a comprehensive review on SARS-CoV-2 entry (Ref. 13).

Two more minor comments:

1. line 526 indicates pixel size for a CCD camera with a tenth of picometer precision which seems rather inflated

We have now removed a significant digit and report a 2.06Å pixel size.

2. The schematic of the EM data processing (Supp Fig. 1) is lacking some statistical/numerical data processing information: how many micrographs were analysed, how many particles went into each classification etc. etc.

The revised version of Supplementary Figure 1 now includes a breakdown of the number of particles analyzed and the number of classes analyzed.